# A novel preparation method for monodisperse streptavidin magnetic beads and its application in electrochemiluminescent immunoassay for CEA

Hengbo Lei[1,2], Yuguo Tang [1,2*]

1 Division of Life Sciences and Medicine, University of Science and Technology of China, Hefei, China,
2 Key Lab of Bio-Medical Diagnostics Chinese Academy of Sciences, Suzhou Institute of Biomedical Engineering and Technology, Suzhou, China

* tangyg@sibet.ac.cn

## Abstract

The magnetic microspheres coated with streptavidin (PS-SA) were successfully prepared by loading streptavidin onto modified hyperbranched poly(glycerol) (HPG) magnetic microspheres. This material was comprehensively characterized using various methods, including scanning electron microscopy, thermogravimetric analysis, fourier-transform infrared spectroscopy, vibrating sample magnetometry and zeta potential measurement. This study introduced a novel electrochemiluminescence immunoassay utilizing functionalized magnetic microspheres for the ultrasensitive detection of carcinoembryonic antigen (CEA). The sensor employed $Ru(bpy)_3^{2+}$-labeled anti-CEA-Ab2 (Ru-Ab2) as the signal probe and combined it with monodisperse streptavidin magnetic beads and biotinylated anti-CEA-Ab1 (bio-Ab1) to selectively capture CEA. Under optimized conditions, the immunosensor exhibited a broad linear range from 0.05 to 1000 ng/mL for CEA detection ($R^2 = 0.998$), along with an exceptionally low detection limit (LOD) of 0.02383 ng/mL. Moreover, the developed immunosensor demonstrated remarkable selectivity, excellent stability, and satisfactory reproducibility, indicating its significant potential for clinical diagnostic applications.

## Introduction

Serum tumor biomarkers play a crucial role in the early detection of cancer and the monitoring of therapeutic responses [1]. Among these biomarkers, carcinoembryonic antigen (CEA), a glycoprotein with a molecular weight of approximately 180–200 kDa, has emerged as one of the most clinically significant markers, particularly for colorectal, gastric, and breast carcinomas [2]. Although healthy adults generally maintain serum CEA levels below 5 ng/mL, malignant diseases—particularly in advanced

**Data availability statement:** All relevant data are within the paper and its Supporting information files. This dataset contains all the data, metadata, and related information necessary to reproduce all the figures and conclusions in the manuscript. We confirm that the minimum dataset required to reproduce the research results of this manuscript has been publicly shared. These data do not involve any ethical or legal restrictions. The original data availability statement continues to apply following the revisions.

**Funding:** This work was supported by Shenzhen Municipal Science and Technology Research and Development Fund (KJZD20240903101100001). The funders had no role in study design, data collection and analysis, decision to publish, or preparation of the manuscript.

**Competing interests:** The authors have declared that no competing interests exist.

stages—can elevate these concentrations to several hundred ng/mL [3]. Serial measurements of CEA provide critical guidance for early clinical intervention and facilitate personalized treatment strategies, thereby revolutionizing paradigms in cancer management.

Recent years have seen significant advancements in the detection technologies for CEA, encompassing a variety of analytical approaches such as fluorescence immunoassays [4,5], colorimetric assays [6], light emitting diode (LED) imaging [7], mass spectrometry-coupled immunoanalysis [8], surface-enhanced raman scattering (SERS) [9], and electrochemical immunosensors [10]. Among these diverse immunoassay techniques, electrochemiluminescence (ECL) stands out as a unique luminescent phenomenon that integrates electrochemical reactions with chemiluminescent processes occurring at or near electrode surfaces [11]. The ECL technique not only offers precise control through electrochemical methods but also possesses high sensitivity in optical signal detection [12]. Unlike photoluminescence techniques, ECL eliminates the need for external excitation sources, thereby mitigating associated background interference [13]. These distinctive characteristics—alongside operational simplicity, an exceptional signal-to-noise ratio, and remarkable sensitivity—have established ECL as a powerful analytical platform with extensive applications across various scientific and technological domains [14].

This study presents an electrochemiluminescence immunosensing strategy based on streptavidin magnetic beads. Compared with mainstream methods, this strategy offers a more convenient and standardized signal amplification approach. Firstly, unlike the traditional methods where probes are directly immobilized on the electrode surface through physical adsorption or covalent bonding – which are usually accompanied by cumbersome electrode pretreatment steps, poor regeneration performance, and difficulty in application to high-throughput detection systems [15] – the magnetic separation platform constructed in this study uses microspheres as mobile solid-phase carriers, effectively reducing the risk of electrode contamination, simplifying the operation process, and achieving efficient enrichment and separation of targets, thus being more suitable for large-scale production [16,17]. Secondly, although the nanomaterial-enhanced sensors developed in recent years can achieve ultra-high sensitivity, they generally have complex synthesis processes, poor batch-to-batch reproducibility, and high costs. In contrast, this method focuses more on good reproducibility, operational stability, and process scalability in its design, which are crucial for practical clinical diagnostic applications. It should be noted that although this method is slightly less sensitive than nanomaterial-enhanced systems [11,16,18–20], metal-organic frameworks (MOFs) [21], or label-free electrochemiluminescence systems [22], in the future, the electrode interface or probes can be modified by introducing these functional materials to further enhance detection sensitivity and automation, thereby promoting the development of advanced point-of-care testing (POCT) technology.

The monodispersity of immunomagnetic beads is crucial for ensuring consistency in bio-recognition and signal reproducibility. However, traditional methods such as suspension polymerization often yield polydisperse microspheres; in contrast,

two-step activated swelling method allows for precise control over particle size and functionality [23]. The surface modification of magnetic beads with hyperbranched poly(glycerol) (HPG) exhibits significant advantages. Compared to linear polymers, hyperbranched poly(glycerol) possesses a higher density of hydrophilic functional groups. The abundant hydroxyl groups confer excellent water dispersibility and enhance the capability for biomolecule capture [24,25]. Furthermore, they exhibit a reduction in nonspecific adsorption that is comparable to that achieved with polyethylene glycol modifications [26], thereby significantly decreasing background noise and reducing detection limits. The conversion of hydroxyl groups to tosyl groups is relatively straightforward, and subsequent reactions with proteins can be facilitated under mild conditions without the need for any coupling agents [27].

As illustrated in Fig 1, the preparation process of PS-SA involves three key steps: First, PS-M was fabricated through a two-step activated swelling process, followed by nitration and magnetization. Subsequently, HPG was grafted onto the surface of PS via surface-initiated anionic ring-opening polymerization to obtain PS-HPG. Finally, hydroxyl groups were reacted with tosyl chloride to convert them into tosyl groups, resulting in PS-tosyl. The obtained PS-tosyl can readily react with amino-containing proteins; in this study, it reacted with streptavidin to produce PS-SA microspheres. Due to its strong binding affinity with biotinylated molecules, streptavidin-coated microspheres serve as a significant biosensing platform. In this study, we developed a sandwich-type immunosensor for the detection of CEA, employing Ru-Ab2, bio-Ab1 and PS-SA magnetic beads. The bio-Ab1 and Ru-Ab2 antibodies bind to the CEA antigen, forming an immune complex. The PS-SA captures this immune complex through high-affinity biotin-streptavidin interactions, thereby completing the sandwich immunoassay. This straightforward and cost-effective electrochemiluminescent immunoassay method facilitates sensitive and reproducible quantitative measurement of CEA within a concentration range of 0.05 to 1000 ng/mL, achieving a detection limit as low as 0.02383 ng/mL. Consequently, the detection range for CEA is exceptionally broad, accompanied by an extremely low detection limit. Furthermore, this ECL immunoassay system is capable of functioning without interference from artificial serum samples, demonstrating its practical applicability for detection purposes.

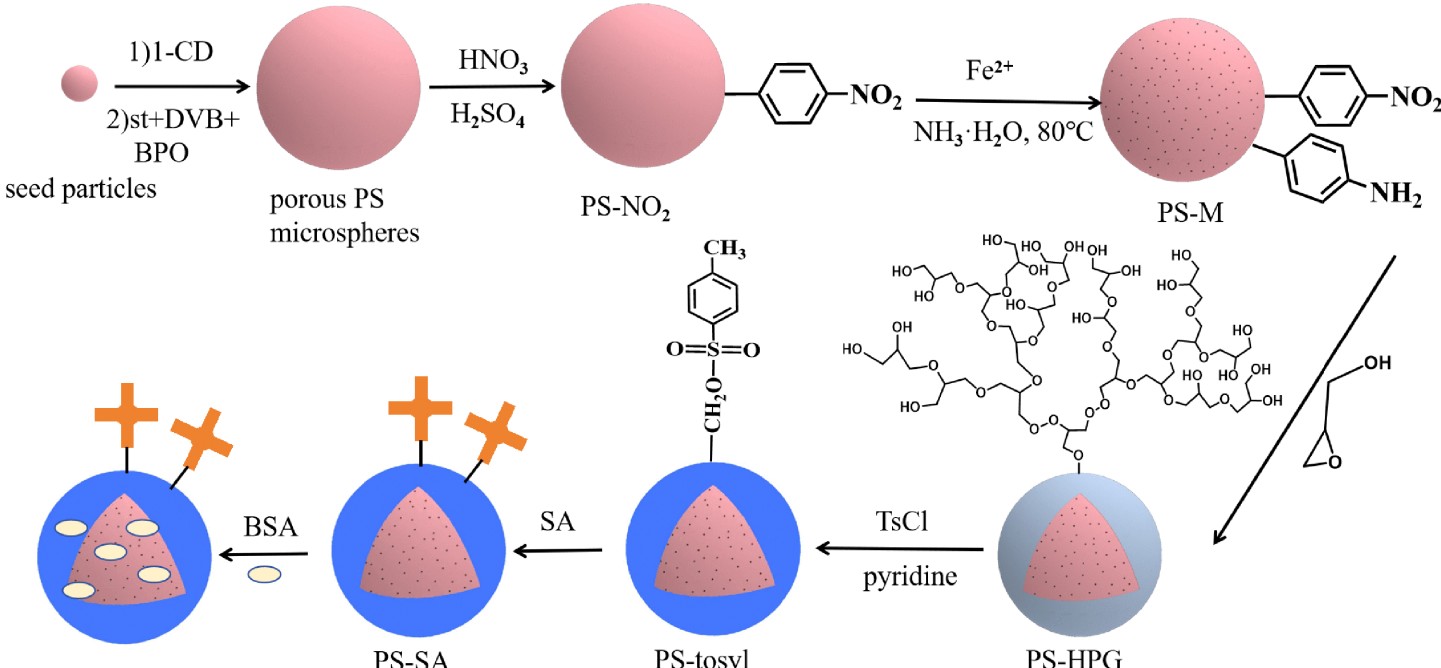

**Fig 1. The preparation process of PS-SA.**

## Experimental

### Chemicals and reagents

Tetrahydrofuran (THF), toluene, polyvinyl alcohol (PVA), diglyme, $FeCl_3 \cdot 6H_2O$, $FeSO_4 \cdot 7H_2O$, tetrahydrofuran (THF), tri-ethylamine, sodium dodecyl sulfate (SDS), benzoyl peroxide (BPO), 1-chlorododecane (1-CD), n-heptane, sulfuric acid, nitric acid, $Na_2HPO_4$, $NaH_2PO_4$, p-toluenesulfonyl chloride (TsCl), acetone, pyridine and aqueous ammonia (25%) were purchased from Sinopharm Group Chemical Reagent Co., Ltd. (Shanghai, China). Sulfo-NHS-LC-LC-Biotin, streptavidin, BSA, EDC, Tween-20, sulfo-NHS, styrene (St) and divinylbenzene (DVB) were purchased from Sigma-Aldrich (St. Louis, MO, USA). The polymerization inhibitor was removed from the St and DVB by alkaline aluminum oxide column chromatography prior to use. Elecsys CEA reagent kit was purchased from F. Hofmann-La Roche & Co. Bis(2,2'-bipyridyl) (4'-methyl- [2']bipyridinyl-4-carboxylicacid)ruthenium(II)dichloride (Ru-COOH) were all purchased from Suna Tech Inc. (Suzhou, China). Streptavidin (SA) were purchased from Bowobiotech Inc. (hangzhou, China). Artificial serum samples were all purchased from Braceds Biotech Co., Ltd. (Shenzhen, China). Thyroid stimulating hormone (TSH), alphafetoprotein (AFP), carbohydrate antigen199 (CA199), prostate specific antigen (PSA), CEA and CEA antibody pair (anti-CEA-ab1 and anti-CEA-ab2) were obtained from Fapon Biotech Co., Ltd. (Guangdong, China).

### Apparatus

ECL measurements were implemented on a Cobas e411 (F. Hofmann-La Roche & Co.). Fourier transform infrared (FTIR) spectra were obtained with the standard KBr disks on a Thermo Scientific Nicolet iS20. Thermogravimetric analysis (TG) was conducted on a NETZSCH TG 209 F1 Tarsus with a heating rate of 10°C/min in nitrogen flow from 50°C to 800°C. Scanning electron microscopy (SEM) images were determined using ZEISS Sigma 300. The zeta potential was assessed using a Malvern Zetasizer Nano. The magnetization curves of microspheres were measured with a LakeShore 7404 Vibrating Sample Magnetometer (VSM) at room temperature.

### Preparation of PS-NO$_2$ and PS-M

The seed microspheres can be obtained from previous experiments, and the detailed preparation method is described in reference [26]. A successful synthesis of monodisperse porous polystyrene microspheres was accomplished through a two-step activation swelling process by Ugelstad [28]. In the initial swelling step, 1.6 g of acetone and 1.2 g of 1-CD were combined with 12 mL of a 0.5 wt% SDS aqueous solution, followed by two rounds of homogenization at a pressure of 400 bar utilizing a high-pressure homogenizer. After the homogenization process, 4 mL of a seed suspension containing 0.4 g of seed microspheres was added, and the activation procedure was maintained at 30°C for a duration of 24 h. In the second step of expansion, a mixture comprising 12 g of St, 10 g of DVB, 10 g of toluene, 8 g of n-heptane, and 0.652 g of BPO was introduced into 240 mL of a 0.25 wt% SDS solution. This mixture underwent two rounds of homogenization using a high-pressure homogenizer set at a pressure of 400 bar before being incorporated into the activated seed particle suspension. After swelling for 2 h at 30°C, 240 mL of 3.2 wt% PVA aqueous solution was added and subsequently heated at 80°C for 24 h under nitrogen atmosphere. The product was collected through repeated centrifugation, followed by the removal of solvents and linear polymers using the soxhlet extraction method with tetrahydrofuran (THF). After multiple washes with methanol and vacuum drying, polystyrene (PS) microspheres were ultimately obtained as the final product.

The synthesis method of PS-NO$_2$, as referenced in Reference [29], has been moderately modified. First, a mixture consisting of 25 g of 95% sulfuric acid and 10 g of 65% nitric acid was cooled to a temperature of 10°C. Subsequently, 1 g of PS microspheres was added to the cooled mixture. The temperature was then increased to 30°C and maintained for 2 h. Following this step, 150 mL of ice water was introduced into the suspension for treatment. Finally, the particles were washed with water and methanol through filtration.

During the preparation of PS-M microspheres, 1 g of PS-NO$_2$ was added to a 20 mL aqueous solution containing 1.2 g of FeSO$_4$·7H$_2$O at room temperature, and then heated to 80°C after stirring for 30 minutes. Then, 15 g of 25% aqueous ammonia solution was added, and the reaction lasted for 1 hour. After cooling, the product was washed several times with methanol and finally dried under vacuum conditions to yield PS-M.

## Synthesis of PS-HPG and PS-SA

According to the previously published procedure [26], PS-M was modified with HPG, resulting in PS-HPG. In summary, 1 g of PS-M was mixed with 500 µL of triethylamine and 30 mL of diglyme, followed by stirring at 90°C for 1 h. Subsequently, 5 g of glycerol was added dropwise, and the reaction continued for an additional 20 h. Finally, the product was washed several times with methanol and dried under vacuum conditions to yield PS-HPG.

The modification process of tosylation was conducted using classical methods, with appropriate adjustments made to the existing procedures. A total of 0.1 g of PS-HPG was reacted with 0.1 g of TsCl and 0.12 mL of pyridine in anhydrous acetone for 24 h at room temperature. The samples were sequentially washed with acetone, a 25% acetone solution, a 50% acetone aqueous solution, and pure water several times. Subsequently, they were preserved in the aqueous phase to obtain a suspension of PS-tosyl at a concentration of 10 mg/mL. A volume of 1 mL of PS-tosyl suspension, pre-washed multiple times with PBS buffer (pH 9.5), was dispersed in 400 µL of PBS buffer (pH 9.5) and subsequently mixed with 290 µL of 3 M ammonium sulfate solution (pH 9.5), followed by incubation at 37°C for 36 hours. The resulting product was designated as PS-SA. Finally, 5% BSA is used as blocking solution to minimize non-specific adsorption. The sample was stored in 0.1M PBS buffer with a pH of 7.4 (containing 0.1% Proclin) at 4°C.

## Preparation of bio-Ab1 and Ru-Ab2

bio-Ab1: Anti-CEA-Ab1 was biotinylated using Sulfo-NHS-LC-LC-Biotin at a 20:1 molar ratio of biotin to antibody. Briefly, 200 µL of the antibody (0.83 µM in PBS) was reacted with 1.66 µL of 2 mM Sulfo-NHS-LC-LC-Biotin (in DMSO) for one hour at room temperature with constant agitation. The reaction mixture was then subjected to buffer exchange and purification via a Zeba spin desalting column (40K MWCO, PBS-equilibrated) to eliminate unconjugated biotin. The purified biotinylated antibody (bio-Ab1) was stabilized in PBS containing 0.1% w/v BSA and stored at 4°C.

Ru-Ab2: Conjugation of the Ru-COOH complex to anti-CEA-Ab2 was performed according to established methods. The Ru-COOH complex (0.83 µL, 10 mM) was activated by reaction with EDC (10 µL, 10 mg/mL) and sulfo-NHS (10 µL, 10 mg/mL) in 30 mM MES buffer (pH 6.0) for 30 minutes at room temperature, yielding the Ru-NHS ester. The activated ester was then added to a solution of anti-CEA-Ab2 (200 µL, 0.83 µM in PBS) at a ruthenium-to-antibody molar ratio of 50:1. The mixture was incubated for 2 hours at 37°C. The resulting conjugate was purified via a Zeba spin desalting column (40K MWCO) equilibrated with PBS to remove unconjugated reactants. The purified Ru-Ab2 was stored in PBS containing 0.1% w/v BSA at 4°C.

The homemade ECL reagent kit consists of the following components: 1) 1 µg/mL bio-Ab1; 2) 1 µg/mL Ru-Ab2; 3) 0.72 mg/mL PS-SA.

## Fabrication of the ECL sensor and electrochemical detection

The preparation process of the immunosensor is illustrated in Fig 2b. The measurement protocol followed the established operational procedures of the instrument. A mixture consisting of 50 µL Ru-Ab2 solution, 50 µL bio-Ab1 solution, and 10 µL PBS (0.01 M, pH 7.4) containing different concentrations of CEA antigen was incubated at 37°C for 9 minutes. Subsequently, the antigen-antibody complex was further incubated with 50 µL PS-SA solution for an additional 9 minutes to construct a sandwich-type immunosensor. To remove physically adsorbed antigens and unbound Ru-Ab2, the resulting product was gently washed with PBS and separated using a magnet. Finally, the immunosensor materials were re-dispersed in 100 µL PBS solution and transferred to the measurement cell where they were magnetically adsorbed onto

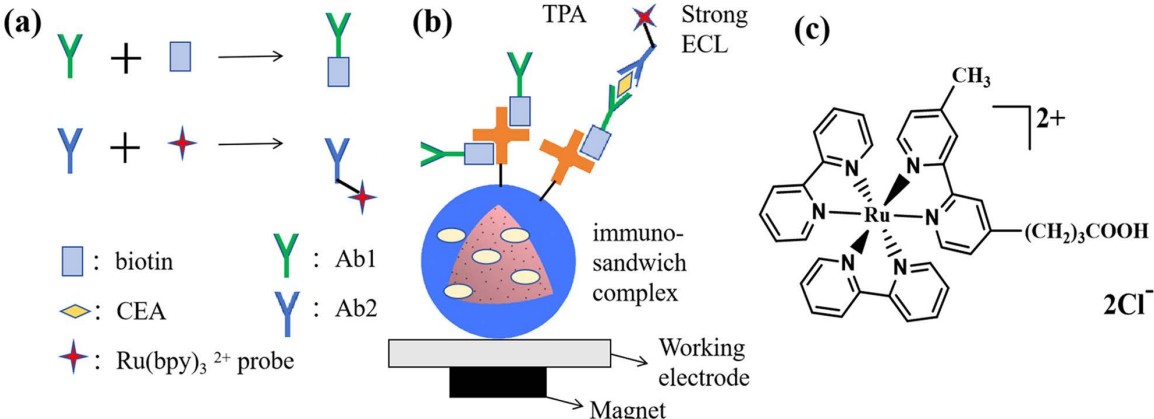

**Fig 2. Schematic diagram of (a) the synthesis of bio-Ab1 and Ru-Ab2; (b) immuno-sandwich complex (PS-SA/bio-Ab1/CEA/Ru-Ab2) and (c) molecular structures of the Ru-COOH dyes.**

the surface of the working electrode. Lastly, an addition of 120 μL TPA solution (180 mM) was made as a co-reactant for ECL testing.

## Results and discussion

### Synthesis and characterization PS-SA

The molecular structure of the probe is illustrated in Fig 2c, where the carboxyl groups are capable of reacting with amino groups on anti-CEA-Ab2, thereby anchoring onto the surface of anti-CEA-Ab2. Fig 2b presents a schematic diagram depicting the arrangement of the sandwich-type electrochemical sensor (PS-SA/bio-Ab1/CEA/Ru-Ab2) on the electrode. Utilizing the superparamagnetic characteristics of magnetic particles, the formed PS-SA/bio-Ab1/CEA/Ru-Ab2 can be rapidly adsorbed onto the surface of the working electrode through magnetic force, and then electrochemiluminescence detection can be carried out.

The SEM images of PS seed particles, PS-NO$_2$, PS-M, PS-HPG, and PS-SA are presented in Fig 3. It can be observed that all these samples exhibit a uniform spherical morphology with consistent particle sizes. Among them, the average diameters of the PS seed particles, PS-NO$_2$, and PS-HPG are measured to be 560 nm, 2.65 μm, and 2.95 μm respectively. After undergoing a two-step activation swelling process, the particle size increased to 4.7 times that of the PS seed particles. Upon completion of the magnetic operations, the magnetic nanoparticles aggregated and clumped on the surface of the spheres, resulting in a further increase in particle diameter. The morphology of PS-M and PS-HPG was observed using SEM (see Fig 3d and 3e). It can be noted that the presence of Fe$_3$O$_4$ nanoparticles leads to partial blockage of pores, resulting in a reduced apparent pore quantity in PS-HPG. At the same time, it can be clearly observed that the surface morphology of PS-HPG has become significantly smoother. This phenomenon may be attributed to the coverage of the surface by polymers or the detachment of unstable magnetic particles during the reaction process. Compared to PS-HPG, the morphologys and sizes of PS-SA exhibit no significant changes (see Fig 3e and 3f).

The typical infrared spectrum of polystyrene is presented in Fig 4a. The asymmetric and symmetric stretching vibrations of the methylene group (-CH$_2$-) are observed at 2923 cm$^{-1}$ and 2853 cm$^{-1}$, respectively. Additionally, the ring vibrations of the benzene ring manifest in absorption bands at 1603 cm$^{-1}$, 1489 cm$^{-1}$, and 1448 cm$^{-1}$. Furthermore, the out-of-plane rotational vibrations of hydrogen atoms on a monosubstituted benzene ring occur at absorption bands of 906 cm$^{-1}$, 759 cm$^{-1}$, and 700 cm$^{-1}$ [30]. Therefore, the presence of polystyrene can be confirmed through these absorption

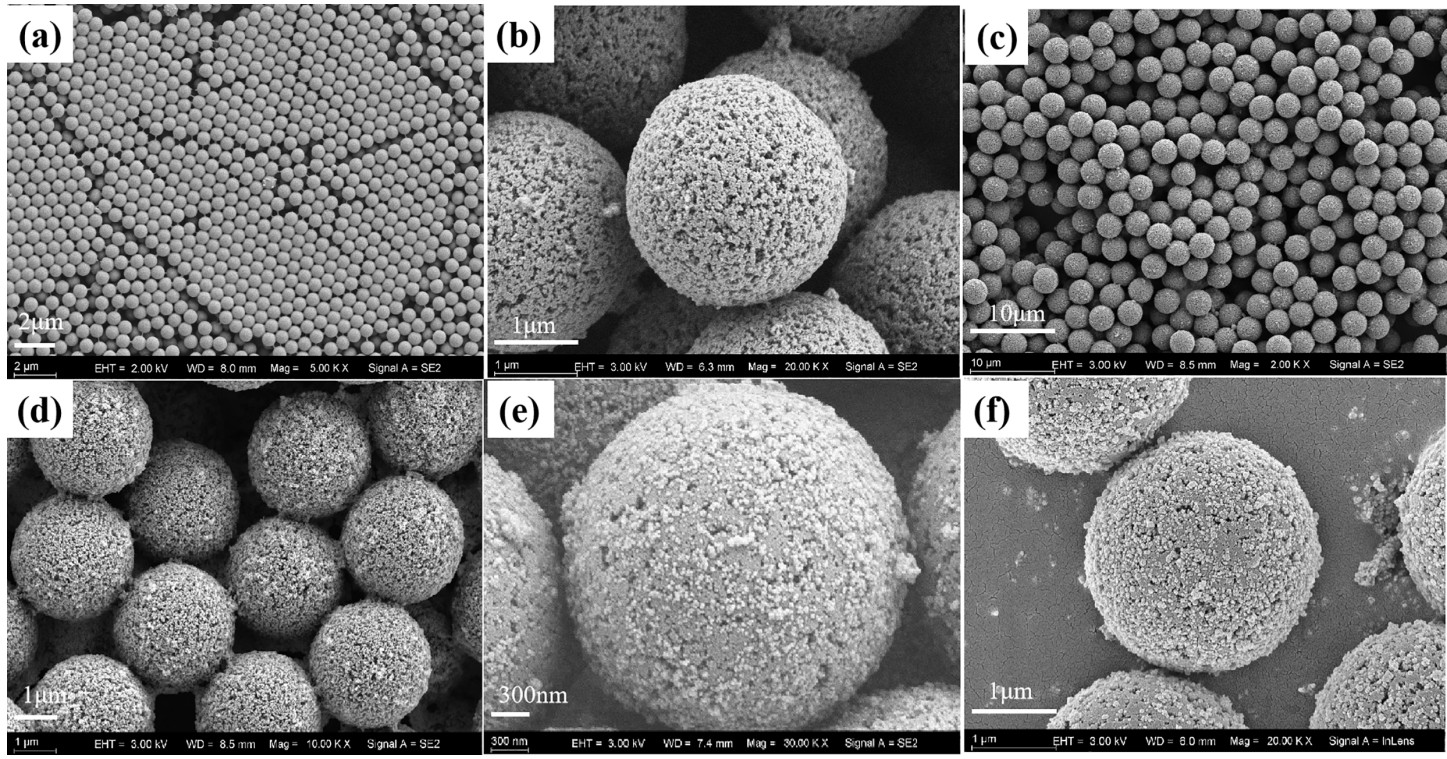

**Fig 3. Low magnification SEM images of (a) PS seed particles; (c) PS-M; high magnification SEM images of (b) porous PS-NO$_2$; (d) PS-M; (e) PS-HPG; (f) PS-SA.**

peaks. After nitration treatment, the infrared spectrum of PS-NO$_2$ exhibits significant absorption peaks at 1526 cm$^{-1}$ and 1349 cm$^{-1}$, which correspond to the stretching vibrations of N-O bonds [29]. This indicates that the nitration process has been successfully implemented. Following modification with ferrous oxide, an absorption band observed at 587 cm$^{-1}$ is identified as the stretching vibration of iron-oxygen bonds [31], further confirming the formation of iron oxide nanoparticles within the porous spheres. In addition, an absorption peak corresponding to the N-H bending vibration is observed at 1663 cm$^{-1}$, indicating that a portion of the -NO$_2$ groups has been reduced to -NH$_2$. In the FTIR spectrum of PS-HPG, a peak at 1055 cm$^{-1}$ is detected, which is attributed to the stretching vibration of the C-O bond and primarily results from an increase in the content of secondary and tertiary alcohols.

After tosylation treatment, the FTIR spectroscopy (Fig 4e) reveals a significant increase in the intensity of the -OH and C-O vibrational absorption peaks at 3413 cm$^{-1}$, 1030 cm$^{-1}$, and 1210 cm$^{-1}$. Following grafting with SA, the peak observed at 1727 cm$^{-1}$ is likely attributed to carboxyl groups and also represents the stretching vibration of C=O [32].

Fig 5a presents the TG curves of PS-M and PS-SA. The primary decomposition range of PS-M is observed between 200°C and 800°C, with nearly complete degradation of organic components by 800°C. Below 800°C, the weight loss rate of PS-M is recorded at 45.3%. In PS-M, the mass fraction of Fe$_3$O$_4$ is approximately 54.7%. Following modification, the proportion of iron cores in the PS-SA microspheres decreases from 54.7% to 47%. The magnetic properties of PS-M and PS-SA are evaluated using VSM. As illustrated in Fig 5b, the saturation magnetization intensities of PS-M and PS-SA reached 15.3 emu/g and 10.4 emu/g, respectively, exhibiting superparamagnetism at room temperature. Following a series of modifications, the saturation magnetization intensity of PS-SA significantly decrease. This phenomenon is attributed to the detachment of loosely adhered magnetic particles from the surface, as well as a reduction in the

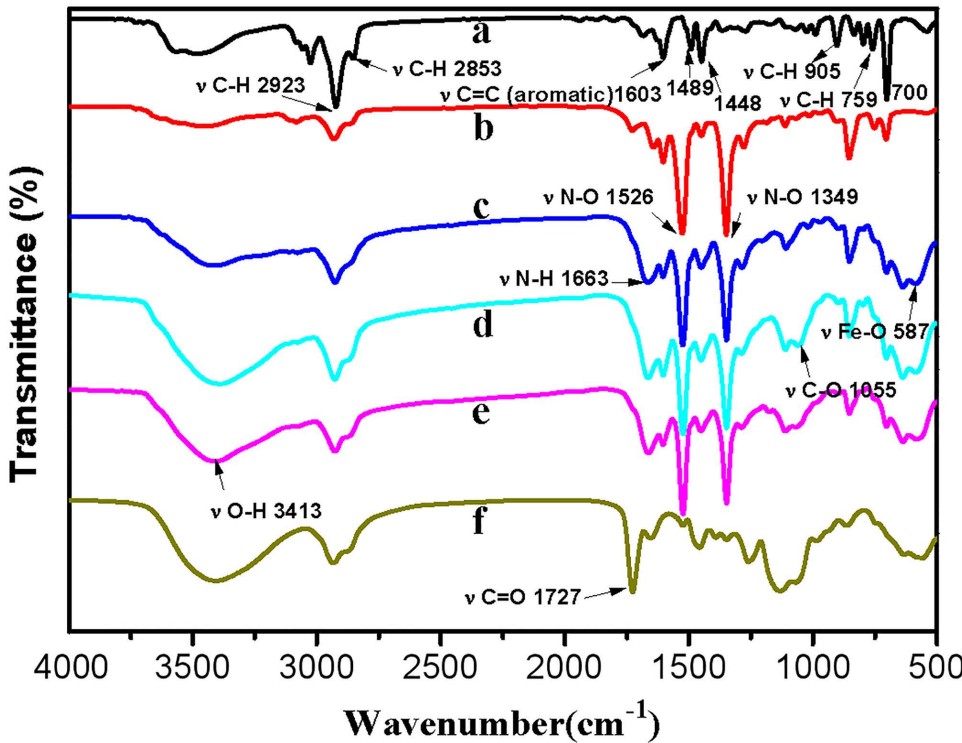

**Fig 4. FTIR spectra curves of (a) porous PS microspheres, (b) PS-NO$_2$, (c) PS-M, (d) PS-HPG, (e) PS-tosyl and (f) PS-SA.**

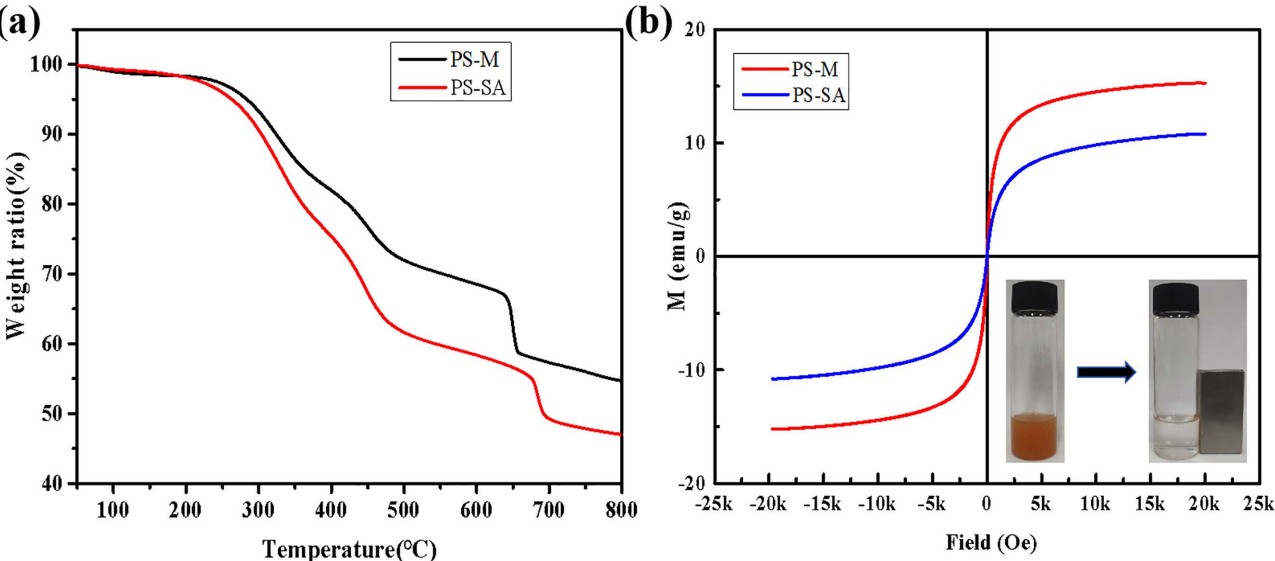

**Fig 5. (a) TG curves of PS-M and PS-SA; (b) Magnetization curves of PS-M and PS-SA measured at 298K.** Insets: the photographs of PS-SA before and afer magnetic separation by an extemal magnet.

proportion of magnetic cores within the microspheres. PS-SA can be easily separated from aqueous solutions by applying an external magnetic field.

The zeta potential data is illustrated in Fig 6. Prior to modification, porous PS microspheres exhibits hydrophobicity with a zeta potential of −7.2 mV. Following nitration modification, polar groups (such as $-NO_2$ and $-SO_4$) are introduced, which enhanced the negative charge on the surface. The addition of magnetic materials increases the mass of the samples, facilitating their sedimentation and thereby reducing the overall tendency for particle dispersion. This modification significantly enhances the zeta potential. After HPG modification, the increase in surface hydroxyl groups enhances the hydrophilicity of the microspheres, further amplifying the negative value of zeta potential. During the tosylation process, however, there is a decrease in hydroxyl group quantity coupled with an increase in toluenesulfonyl group quantity, this leads to diminished hydrophilicity and weakened electronegativity. Upon completion of SA grafting, due to the abundance of carboxyl groups on SA protein, this change further augment hydrophilicity and renders zeta potential even more negative. These variations in zeta potential clearly demonstrate the successful implementation of each modification step.

## Detection of CEA

A standard calibration curve for CEA detection in PBS has been established. The concentrations of CEA measured were 0.05, 0.1, 0.2, 0.5, 1, 10, 30, 100, 200, 300, 500, 700, and 1000 ng/mL. Three replicate experiments were conducted at each concentration level. As the concentration of CEA increased, the ECL intensity gradually enhanced. Furthermore, the relative standard deviation (RSD) at different concentrations was consistently below 5%, indicating that the ECL measurements exhibit good stability. Within the concentration range of 0.05 to 1000 ng/mL, a linear relationship between ECL intensity and CEA concentration was observed. This linear relationship can be expressed by the following equation: $I = 550.67c + 1265.31$, where I represents the ECL intensity and c denotes the CEA concentration (ng/mL), with a correlation coefficient $R^2 = 0.998$. Fig 7b indicates that even within lower concentration intervals, there remains a strong linear correlation between the two variables, without any notable deviation from the overall fitting curve results. Therefore, a unified linear relationship can be employed to analyze data spanning a wide range of concentrations.

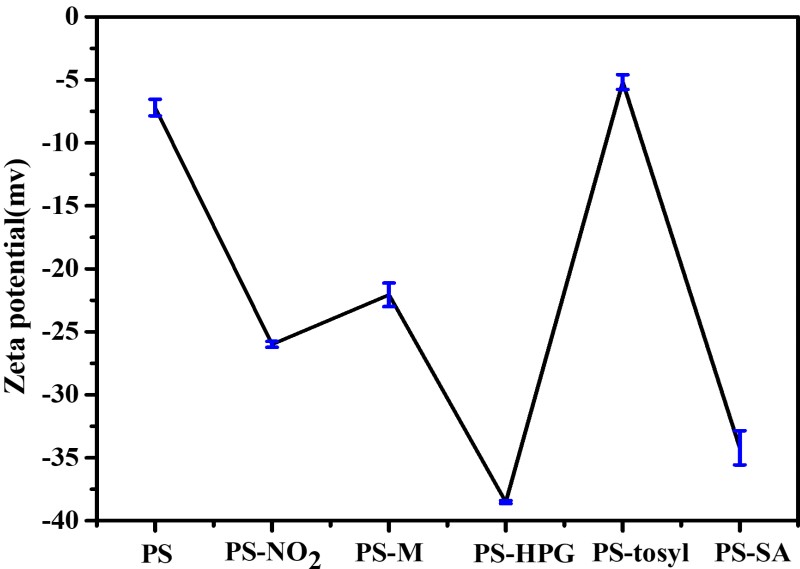

**Fig 6. Zeta potential results of porous PS microspheres, PS-NO$_2$, PS-M, PS-HPG, PS-tosyl and PS-SA.**

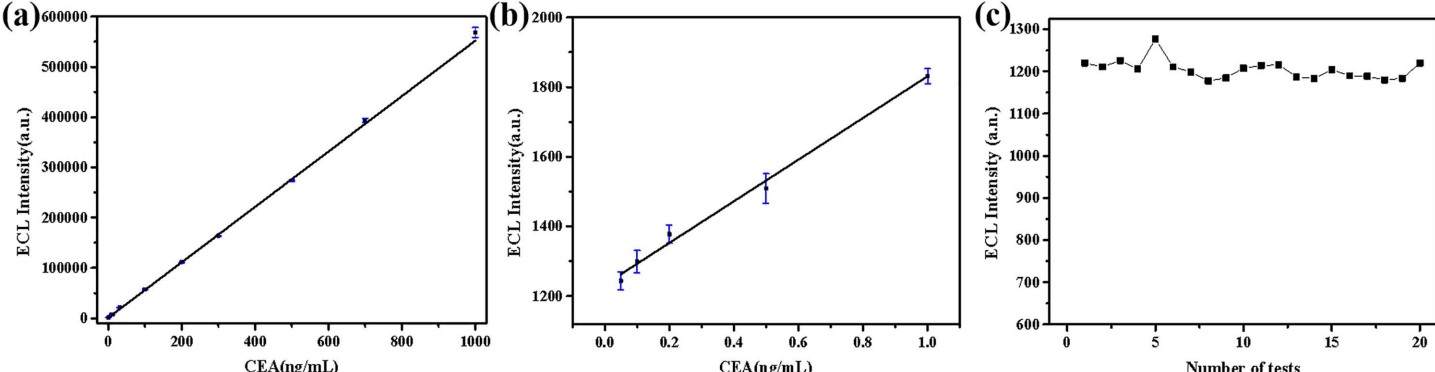

**Fig 7. Calibration curve of ECL intensity (a) full range concentration and (b) low concentrations of CEA (ng/mL).** (c) The blank sample was measured repeatedly 20 times.

The blank PBS solution was subjected to 20 repeated measurements, and the mean value (CM) and standard deviation (SD) were calculated. Relevant data can be found in Fig 7c and Table 1. The formula for determining the low detection limit is expressed as follows: LOD = 3SD/k (k = slope of the calibration plot). The low detection limit is established at 0.02383 ng/mL. Compared with the Roche assay, which is considered the gold standard for tumor marker detection, the proposed method exhibits superior analytical performance. According to the manufacturer's specifications, the Roche kit has a linear measurement range of 0.3–1000 ng/mL and a LOD of 0.269 ng/mL. In contrast, the developed method achieves a broader linear range and demonstrates an LOD of 0.02383 ng/mL, representing a tenfold improvement in sensitivity.

As shown in Table 2, when the linear range and detection limit of the reported CEA determination methods are compared with the corresponding indicators of the method developed in this study, the results indicate that this method outperforms the vast majority of existing methods in terms of the detection range. Although the detection limit obtained has not yet reached the highest level reported for sensing platforms based on MOFs, it is sufficient to meet the clinical detection requirements of the target analyte when only 10 µL of sample is used.

**Table 1. The mean value (CM) and standard deviation (SD) of the blank sample measured repeatedly 20 times.**

| Sample | CM | SD |
|---|---|---|
| PBS buffer solution | 1204.45 | 14.37 |

**Table 2. The comparison of the linear range and detection limit reported method with that of the developed method for the determination of CEA.**

| Samples | Linear range (ng/mL) | LOD (ng/mL) | Ref |
|---|---|---|---|
| ACN-sensitized P5ICA | 0.01-500 | 0.0033 | [2] |
| Au-β-CD/MXene@PANI/FTO | 0.05-350 | 0.429 | [33] |
| Anti-CEA/PBSE/graphene/Cu | 1-25 | 0.23 | [34] |
| BSA/antiCEA/ZnO/HGO/GCE | 0.1-20 | 0.07 | [35] |
| Zn-TBAPy/ZIF-67@PDA | $1 \times 10^{-4}$-80 | $2.75 \times 10^{-4}$ | [21] |
| Ru@Zn-MOF-74/SiO2@PDA | $1 \times 10^{-4}$-80 | $1.26 \times 10^{-5}$ | [36] |
| PS-SA/bio-Ab1/CEA/Ru-Ab2 | 0.05-1000 | 0.02383 | This work |

## Selectivity, stability, and reproducibility of the sensor

In order to validate the feasibility of the proposed sensor for detecting actual samples, it is essential to conduct a comprehensive study on its specificity towards CEA. We evaluated the sensor's response to other tumor markers and hormones, including AFP, CA199, TSH, and PSA. In the selective experiments, the concentration of CEA was set at 30 ng/mL, while the concentrations of other interfering proteins were maintained at elevated levels (see Fig 8A). The results indicated that

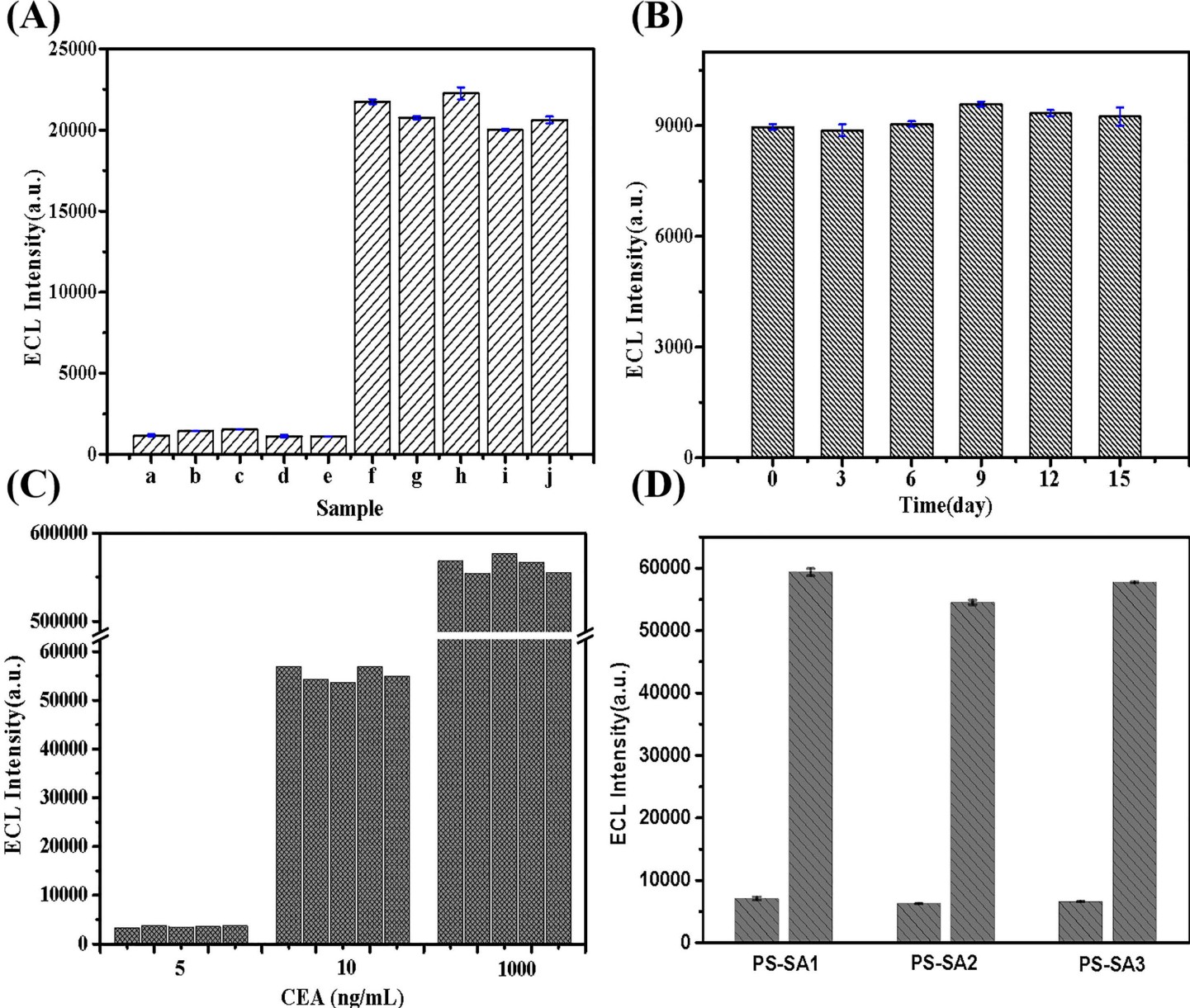

**Fig 8. (A) Specificity investigations of the immunosensor for CEA detection.** (a) blank, (b) AFP (300 ng/mL), (c) CA199 (100 U/mL), (d) TSH (50 uIU/mL), (e) PSA (50 ng/mL), (f) CEA (30 ng/mL), (g) CEA (30 ng/mL) + AFP (300 ng/mL), (h) CEA (30 ng/mL) + CA199 (100 U/mL), (i) CEA (30 ng/mL) + TSH (50 uIU/mL), (j) CEA (30 ng/mL) + PSA (50 ng/mL). (B) Thermostability of the PS-SA beads for the detection at 15 ng/mL of CEA. (C) Reproducibility studies of the immunosensor at 5, 100 and 1000 ng/mL of CEA. (D) The batch-to-batch differences in the electrochemiluminescence detection results among the three independent batches of magnetic beads (PS-SA1, PS-SA2, and PS-SA3).

only a slight electrochemiluminescence response was observed when detecting the selected tumor markers or hormones. Notably, the electrochemiluminescence response for CEA was significantly higher than that for other types of proteins. Furthermore, when assessing mixtures of various tumor markers or hormones with CEA, the luminescence intensity generated by these mixed solutions was comparable to that measured from individual CEA solutions. This finding suggests that these interfering substances did not adversely affect the detection of CEA. These results demonstrate that this immunosensor exhibits good selectivity in detecting CEA.

The thermal stability of PS-SA microspheres is of significant importance in analytical applications. To evaluate the stability of the microspheres, accelerated aging was conducted at 37°C, with periodic testing of their ECL signals. As shown in Fig 8B, a relatively stable ECL curve was obtained at a concentration of 15 ng/mL. After storage at 37°C for 15 days, the ECL intensity of the biosensor remained within an acceptable range, with a relative deviation of only 8% from its initial value. These results indicate that this ECL immunosensor exhibits excellent stability. Despite this promising result, It should be noted that the results of accelerated stability studies cannot be directly extrapolated to predict long-term stability under real-world storage conditions. Consequently, investigating the long-term stability of key components, such as the ruthenium complex and the antibody itself, remains a critical and valuable direction for future research.

The three concentration values of low, medium, and high (5 ng/mL, 100 ng/mL, and 1000 ng/mL) were selected for five repeated measurements to evaluate their reproducibility. The relative standard deviations (RSD) obtained for the high, medium, and low concentration values were 5.5%, 2.8%, and 1.7%, respectively (see Fig 8C). These results indicate that the ECL immunosensor exhibits good reproducibility. It can be inferred that for detecting exceptionally low concentrations, such as 0.1 ng/mL, the value is often at or near the assay's lower LOD. At this concentration level, the CV typically increases significantly, indicating reduced measurement precision. This undoubtedly presents a clear technical challenge and drives the demand for developing novel electrochemical sensors with superior sensitivity and precision.

To assess the reproducibility of the preparation process, three independent batches of streptavidin-functionalized magnetic beads (PS-SA1, PS-SA2, and PS-SA3) were produced under identical conditions. SEM revealed consistent particle size distribution across all batches (CV < 5%; S1 Fig). Electrochemiluminescence analysis further demonstrated minimal batch-to-batch variation, with response differences below 6.2% (Fig 8D). These findings confirm the high reproducibility of the fabrication process. The two-step activated swelling method demonstrates notable advantages for large-scale production, particularly in terms of superior scalability, high reproducibility, and improved cost-efficiency. This approach ensures consistent particle size distribution across batches and contributes to a more streamlined manufacturing process.

## Interference of artificial serum

The artificial serum was used as a blank control, and the standard addition method was employed for the study. In the aforementioned experiments, artificial serum replaced PBS buffer solution. By adding a specific amount of CEA antigen, simulated positive serum samples were prepared. The diluted artificial serum samples, along with those diluted in PBS buffer, were subjected to ECL experiments under identical conditions. The ECL intensities of both sample types were compared to assess the interference effect of artificial serum on the detection results. Fig 9 illustrates the relative ECL intensity.

The results indicate that there is no significant difference in ECL intensity between artificial serum dilution samples and PBS dilution samples. Specifically, the ECL intensity of artificial serum dilution samples compared to PBS dilution samples was 99.81% for high values and 95.91% for low values, respectively. When detecting low concentrations of CEA, the influence of artificial serum can significantly increase; however, it remains within an acceptable range. This suggests that potential interfering factors present in artificial serum do not significantly impact CEA detection. However, it should be noted that although artificial serum provides a controllable matrix for performance evaluation, it cannot fully simulate the complex components and endogenous substances in real human serum that may cause matrix effects and interference. To enhance the clinical applicability of the method, subsequent research will systematically evaluate the role of dedicated

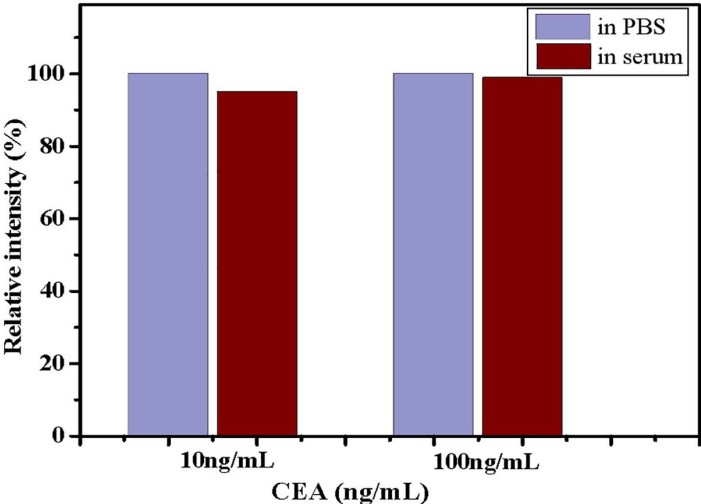

**Fig 9. The impact of different matrices (artificial serum and PBS) on ECL intensity.**

blockers and conduct clinical validation experiments based on the serum of patients and healthy individuals to comprehensively examine the diagnostic accuracy, recovery rate, and matrix interference resistance of the method, ultimately ensuring its effectiveness and robustness in clinical application.

Furthermore, the detection results of simulated artificial serum samples using homemade reagents were compared with those obtained from the Elecsys CEA reagent kit, employing a standardized electrochemiluminescence immunoassay protocol. The results reported in Table 3 demonstrate that the immunosensor developed based on PS-SA/bio-Ab1/CEA/Ru-Ab2 exhibits good reliability in measuring CEA. It is noteworthy that while the cost of our in-house developed reagent kit has been significantly reduced compared to the commercially available Elecsys CEA reagent kit, which is notably expensive, the expenses associated with conjugation reagents and antibody raw materials remain substantial in the context of industrial-scale application. Future work will prioritize the exploration of more cost-effective alternatives and the scaling-up of the synthesis process, with the aim of assessing its industrial feasibility and economic sustainability.

## Conclusion

We successfully prepared microspheres with surfaces rich in hyperbranched hydroxyl polymers (HPG) using seed swelling polymerization technology. These abundant surface groups provide effective anchoring points for the immobilization of biomolecules. Subsequently, we employed a straightforward method to modify the hydroxyl groups through tosylation, enabling the chemical conjugation of SA and resulting in the synthesis of monodisperse magnetic beads. Due to the specific interactions between antigens and antibodies, well-dispersed PS-SA was synthesized as a magnetically separable carrier for targeting CEA. This system is capable of forming a sandwich-type structure comprising PS-SA/bio-Ab1/CEA/Ru-Ab2.

**Table 3. Comparison of detection results from simulated artificial serum samples using homemade reagents and Elecsys CEA reagent kit.**

| Sample | Homemade reagents (ng/mL) | Elecsys CEA (ng/mL) |
| --- | --- | --- |
| 1 | 1.73 | 1.77 |
| 2 | 0.89 | 0.91 |
| 3 | 3.45 | 3.46 |

The electrochemical platform achieved an extensive detection range (0.05–1000ng/mL) for CEA antigens, demonstrating a sensitivity of 0.02383 ng/mL, which surpasses that of traditional methods. Consequently, this immunosensor has been successfully applied in CEA detection, exhibiting satisfactory sensitivity, stability, reproducibility, and selectivity. Finally, the ruthenium-based sandwich-type electrochemiluminescent immunosensing platform we proposed demonstrates significant practical application potential in the analysis of serum samples.

## Supporting information

**S1 Fig. The differences in the SEM images of three independent batches of magnetic beads (PS-SA1, PS-SA2 and PS-SA3).**
(TIF)

**S1 Table. The various characterization data of synthetic materials.**
(XLSX)

**S2 Table. Data related to zeta potential and electrochemiluminescence detection.**
(XLSX)

## Acknowledgments

This project was made possible by the data collection efforts of many researchers including Xiang Cao, Zhiping Jia and Yadong Li. We also thank the electrochemical platform of the Suzhou Institute of Biomedical Engineering and Technology, Chinese Academy of Sciences for its technical support.

## Author contributions

**Conceptualization:** Yuguo Tang.

**Data curation:** Hengbo Lei.

**Formal analysis:** Hengbo Lei, Yuguo Tang.

**Funding acquisition:** Hengbo Lei.

**Methodology:** Hengbo Lei.

**Project administration:** Yuguo Tang.

**Resources:** Yuguo Tang.

**Software:** Hengbo Lei.

**Supervision:** Hengbo Lei.

**Writing – original draft:** Hengbo Lei.

**Writing – review & editing:** Hengbo Lei.

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
