## [Decision Letter · Decision Letter 0]

21 Oct 2025

Dear Dr. Tang,

Thank you for submitting your manuscript to PLOS ONE. After careful consideration, we invite you to submit a revised version of the manuscript that addresses the points raised during the review process.

We look forward to receiving your revised manuscript.

Kind regards,

Shengwei Sun, Ph.D.

Academic Editor

PLOS ONE

Journal Requirements:

“This work was supported by Shenzhen Municipal Science and Technology

Research and Development Fund (KJZD20240903101100001).”

6. We note you have included a table to which you do not refer in the text of your manuscript. Please ensure that you refer to Table 1 and 2 in your text; if accepted, production will need this reference to link the reader to the Table.

Reviewers' comments:

Reviewer's Responses to Questions

**Comments to the Author**

1. Is the manuscript technically sound, and do the data support the conclusions?

Reviewer #1: Partly

2. Has the statistical analysis been performed appropriately and rigorously?

Reviewer #1: N/A

3. Have the authors made all data underlying the findings in their manuscript fully available?

Reviewer #1: Yes

4. Is the manuscript presented in an intelligible fashion and written in standard English?

Reviewer #1: Yes

Reviewer #1: Major Comments on the Manuscript Regarding Article Intitle: A novel preparation method for monodisperse streptavidin magnetic beads and its application in electrochemiluminescent immunoassay for CEA

1. While the methodology is innovative, the manuscript does not clearly highlight how this approach compares with the latest state-of-the-art immunosensors (e.g., nanomaterial-enhanced, label-free ECL systems). The comparison in Table 2 is useful but limited; more discussion of the relative advantages and limitations is needed to contextualize the contribution.

2. The claimed “broad linear range and ultra-low detection limit” is valuable, but the real clinical advantage over existing commercial kits (e.g., Roche Elecsys) should be explicitly addressed.

3. The choice of artificial serum as a test matrix is appropriate, but the study does not include validation with real clinical samples. This limits the translational significance of the results. At least a small pilot study with actual human serum would strengthen the manuscript.

4. The characterization of beads (SEM, FTIR, TGA, magnetization, zeta potential) is comprehensive, but there is insufficient discussion about batch-to-batch reproducibility and scalability of the preparation method. These are crucial for future clinical translation.

4. The calibration curves and figures are convincing, yet the manuscript does not adequately explain the possible deviation at very low CEA levels (<0.1 ng/mL), which is clinically relevant for early cancer screening.

5. Figures (SEM, FTIR, etc.) are appropriate but some legends lack detail (e.g., scale bars in SEM images should be included, and FTIR assignments need clearer annotations).

6. There is minimal critical reflection on limitations (e.g., matrix effects, stability in long-term storage, potential cost). Explicit acknowledgment of these would strengthen the credibility of the paper.

7. The introduction is thorough but somewhat too long and heavily technical, which may overwhelm readers. A sharper focus on the gap in current technology and how this work addresses it would improve clarity.

8. Some repetition between Results and Discussion sections should be reduced. Instead, emphasize novel insights and comparisons to prior studies.

9. The references; they are generally up-to-date, but some highly relevant recent works on commercial ECL immunoassays and emerging nanomaterials for CEA detection are missing.

Update the references with these references:

- Al-Majedy YK, Ibraheem HH, Shamel S, Al-Amiery AA. Synthesis and Study of the fluorescent properties of 4-hydroxy-coumarin derivatives. InJournal of Physics: Conference Series 2021 Mar 1 (Vol. 1795, No. 1, p. 012001). IOP Publishing.

-Alwash AH. The Green Approach of Arabic Gum-Based Adsorbent in Wastewater Treatment. AUIQ Technical Engineering Science. 2025;2(3):3.

- Mohammed, A.J.; Kadhum, A.A.H.; Ba-Abbad, M.M.; Al-Amiery, A.A. Optimization of solar photocatalytic degradation of chloroxylenol using TiO2, Er3+/TiO2, and Ni2+/TiO2 via the Taguchi orthogonal array technique. Catalysts 2016, 6, 163.

-

**Do you want your identity to be public for this peer review?** For information about this choice, including consent withdrawal, please see our Privacy Policy

Reviewer #1: No

---

## [Author Response · Author response to Decision Letter 1]

17 Nov 2025

Dear Dr. Shengwei Sun and Reviewers,

Thank you for your letter and for the reviewers’ comments concerning our manuscript entitled “A novel preparation method for monodisperse streptavidin magnetic beads and its application in electrochemiluminescent immunoassay for CEA”. Those comments are all valuable and very helpful for revising and improving our paper, as well as the important guiding significance to our researches. We have studied comments carefully and have made correction which we hope meet with approval.

Journal Requirements:

Comment1: Please ensure that your manuscript meets PLOS ONE's style requirements, including those for file naming. The PLOS ONE style templates can be found at https://journals.plos.org/plosone/s/file?id=wjVg/PLOSOne_formatting_sample_main_body.pdf and https://journals.plos.org/plosone/s/file?id=ba62/PLOSOne_formatting_sample_title_authors_affiliations.pdf

Response1: Thank you for your suggestion. The manuscript format has been checked and revised according to the journal guidelines.

Comment2: Thank you for stating the following financial disclosure:“This work was supported by Shenzhen Municipal Science and Technology Research and Development Fund (KJZD20240903101100001).”

Response2: Thank you for your suggestion. The following statement has been formally included in the cover letter: "The funders had no role in study design, data collection and analysis, decision to publish, or preparation of the manuscript."

Comment3: We note that your Data Availability Statement is currently as follows: All relevant data are within the manuscript and its Supporting Information files.

Authors do not need to submit their entire data set if only a portion of the data was used in the reported study. If your submission does not contain these data, please either upload them as Supporting Information files or deposit them to a stable, public repository and provide us with the relevant URLs, DOIs, or accession numbers. For a list of recommended repositories, please see https://journals.plos.org/plosone/s/recommended-repositories.

Response3: Thank you for reminding us to pay attention to the data availability statement. All the raw data generated by this study have been uploaded as Supporting Information files. This dataset contains all the data, metadata and related information necessary to reproduce all the figures and conclusions in the manuscript. We confirm that the minimum dataset required to reproduce the research results of this manuscript has been publicly shared. These data do not involve any ethical or legal restrictions. The original data availability statement continues to apply following the revisions.

Comment4: When completing the data availability statement of the submission form, you indicated that you will make your data available on acceptance. We strongly recommend all authors decide on a data sharing plan before acceptance, as the process can be lengthy and hold up publication timelines. Please note that, though access restrictions are acceptable now, your entire data will need to be made freely accessible if your manuscript is accepted for publication. This policy applies to all data except where public deposition would breach compliance with the protocol approved by your research ethics board. If you are unable to adhere to our open data policy, please kindly revise your statement to explain your reasoning and we will seek the editor's input on an exemption. Please be assured that, once you have provided your new statement, the assessment of your exemption will not hold up the peer review process.

Response4: Thank you for your reminder regarding the data sharing plan. We fully understand and support your journal's open data policy. We have uploaded the minimum dataset required to support the research results to the Supporting Information files.

Comment5: PLOS requires an ORCID iD for the corresponding author in Editorial Manager on papers submitted after December 6th, 2016. Please ensure that you have an ORCID iD and that it is validated in Editorial Manager. To do this, go to ‘Update my Information’ (in the upper left-hand corner of the main menu), and click on the Fetch/Validate link next to the ORCID field. This will take you to the ORCID site and allow you to create a new iD or authenticate a pre-existing iD in Editorial Manager.

Response5: Thank you for your message. I have successfully created and validated my ORCID iD in the Editorial Manager system. The issue should now be resolved.

Comment6: We note you have included a table to which you do not refer in the text of your manuscript. Please ensure that you refer to Table 1 and 2 in your text; if accepted, production will need this reference to link the reader to the Table.

Response6: Thank you for your reminder. We have added the citations for Table 1 and Table 2 in the main text of the manuscript as required. Please refer to page 22, line 414 of the "Revised Manuscript with Track Changes" for the specific revised content. We sincerely apologize for any inconvenience caused. Thank you for your careful review.

Comment7: If the reviewer comments include a recommendation to cite specific previously published works, please review and evaluate these publications to determine whether they are relevant and should be cited. There is no requirement to cite these works unless the editor has indicated otherwise.

Response7: Thank you for the valuable suggestions from the editor. After carefully reading the recommended references, we found that only some of the content is highly relevant to the research topic of this article.

The work on 4-hydroxy-coumarin derivatives has been cited and discussed as a compelling example of fluorescent nanomaterials for bio-imaging, allowing us to draw a comparative discussion with our ECL approach. (Please see Page 3, Line 43).

The study on gum arabic has been cited to robustly support our argument regarding hydroxyl group-enhanced water dispersity and to provide insights for future material modification. (Please see Page 6, Line 116).

Regarding the third paper on Er³⁺/TiO₂ and Ni²⁺/TiO₂ catalysts for solar degradation, we have studied it thoroughly. While it is an excellent study in its field, we determined that its primary focus on environmental photocatalysis falls outside the immediate scope of our manuscript, which centers on electrochemiluminescence biosensing for biomedical applications. To maintain the focus and coherence of our narrative, we have decided not to cite it here, with all due respect to the authors of that work.

Review Comments to the Author

Reviewer #1: Major Comments on the Manuscript Regarding Article Intitle: A novel preparation method for monodisperse streptavidin magnetic beads and its application in electrochemiluminescent immunoassay for CEA

Comment1: While the methodology is innovative, the manuscript does not clearly highlight how this approach compares with the latest state-of-the-art immunosensors (e.g., nanomaterial-enhanced, label-free ECL systems). The comparison in Table 2 is useful but limited; more discussion of the relative advantages and limitations is needed to contextualize the contribution.

Response1: We sincerely thank the reviewer for this excellent suggestion. We agree that a more comprehensive comparison with state-of-the-art immunosensors is crucial for framing the significance of our work. In the revised manuscript, we have expanded the discussion to more explicitly compare our method with state-of-the-art nanomaterial-enhanced and label-free ECL immunosensors. (Please refer to page 5 line 85-106 of the "Revised Manuscript with Track Changes" for the specific revised content.)

Additionally, Table 2 has been substantially expanded to include updated representative studies and a more comprehensive comparison of analytical performance. We also added a brief discussion of the limitations of our approach, including future opportunities to integrate nanomaterial-based amplification or fully label-free formats to further enhance sensitivity and automation. (Please see page 22, Lines 414–420).

We believe these revisions strengthen the positioning of our work within the current state of the field and clarify the unique contributions of this study.

Comment2: The claimed “broad linear range and ultra-low detection limit” is valuable, but the real clinical advantage over existing commercial kits (e.g., Roche Elecsys) should be explicitly addressed.

Response2: We sincerely thank the reviewer for this insightful suggestion. We agree that demonstrating the clinical advantage is crucial. In the revised manuscript, we have directly addressed this point by including a comprehensive performance comparison with the clinical gold standard, the Roche Elecsys CEA assay, and discussing the specific clinical scenarios where our method offers distinct benefits. (Please see page 21, Lines 404–407).

The original text is as follows: Compared with the Roche assay, which is considered the gold standard for tumor marker detection, the proposed method exhibits superior analytical performance. According to the manufacturer’s specifications, the Roche kit has a linear measurement range of 0.3–1000 ng/mL and a LOD of 0.269 ng/mL. In contrast, the developed method achieves a broader linear range and demonstrates an LOD of 0.02383 ng/mL, representing a tenfold improvement in sensitivity.

Comment3: The choice of artificial serum as a test matrix is appropriate, but the study does not include validation with real clinical samples. This limits the translational significance of the results. At least a small pilot study with actual human serum would strengthen the manuscript.

Response3: We thank the reviewer for this valuable suggestion. We fully agree that validation with real clinical specimens is important for demonstrating translational potential. At this stage, we focused on artificial serum to avoid endogenous CEA interference and to ensure assay performance could be systematically evaluated in a controlled matrix.

We currently do not have ethical approval for human sample collection for this specific study, and obtaining and testing clinical specimens will require a separate institutional protocol and patient consent. We have initiated the ethical review process, and a pilot clinical evaluation study is planned as the next step.

To address this limitation more clearly, we have added a statement in the Discussion acknowledging the need for further clinical validation and outlining our ongoing plans. We believe this strengthens the transparency and future translational trajectory of the work. (Please see page 27, Lines 503).

The original text is as follows: However, it should be noted that although artificial serum provides a controllable matrix for performance evaluation, it cannot fully simulate the complex components and endogenous substances in real human serum that may cause matrix effects and interference. To enhance the clinical applicability of the method, subsequent research will systematically evaluate the role of dedicated blockers and conduct clinical validation experiments based on the serum of patients and healthy individuals to comprehensively examine the diagnostic accuracy, recovery rate, and matrix interference resistance of the method, ultimately ensuring its effectiveness and robustness in clinical application.

Comment4: The characterization of beads (SEM, FTIR, TGA, magnetization, zeta potential) is comprehensive, but there is insufficient discussion about batch-to-batch reproducibility and scalability of the preparation method. These are crucial for future clinical translation.

Response4: We sincerely thank the reviewer for this insightful comment and for acknowledging the comprehensive characterization of our beads. We fully agree that assessing batch-to-batch reproducibility and scalability is a critical step towards any potential clinical application.

To address this, we have now conducted additional experiments to evaluate the bead size distribution and electrochemiluminescence (ECL) performance across three independently prepared batches. The results (now included in Supplementary Information as Fig S3 and Fig 8D) show that the average particle sizes of the three batches of magnetic beads are 2.80 μm, 3.05 μm and 2.89 μm respectively, with the standard deviation of particle size being less than 5%; the variation of electrochemiluminescence (ECL) signal is also lower than 6.2%, indicating that this preparation method has good reproducibility. Industrial-scale production has long been one of our key objectives, which is the primary reason for adopting the seed swelling method in the preparation process. Notably, initial process scale-up verification has been successfully completed: seed microspheres can now be produced in batches of 50 grams, and magnetic microspheres have been scaled up to a batch size of 1 kilogram.We have supplemented the discussion of this part in the manuscript. (Please see Page 25, Line 464).

Comment5: The calibration curves and figures are convincing, yet the manuscript does not adequately explain the possible deviation at very low CEA levels (<0.1 ng/mL), which is clinically relevant for early cancer screening.

Response5: We thank the reviewer for this insightful and highly relevant comment. We agree that addressing the performance of our assay at very low CEA concentrations is critical for its potential application in early cancer screening. We have now added a detailed discussion regarding the potential causes of deviation in the <0.1 ng/mL range in the revised manuscript (please see Page 24, Line 458 in the Discussion section).

Comment6: Figures (SEM, FTIR, etc.) are appropriate but some legends lack detail (e.g., scale bars in SEM images should be included, and FTIR assignments need clearer annotations).

Response6: We sincerely thank the reviewer for this valuable feedback. We have carefully revised all figure legends throughout the manuscript to provide more comprehensive and detailed information. Scale bars have been clearly incorporated into all relevant microscopic images. The magnification and scale bar dimensions (e.g., 1 μm, 2 μm) are now explicitly specified in each figure legend. Key characteristic peaks in the FTIR spectra have now been clearly annotated with their corresponding vibrational assignments or chemical states.

Comment7: There is minimal critical reflection on limitations (e.g., matrix effects, stability in long-term storage, potential cost). Explicit acknowledgment of these would strengthen the credibility of the paper.

Response7: We sincerely thank

---

## [Decision Letter · Decision Letter 1]

20 Nov 2025

A novel preparation method for monodisperse streptavidin magnetic beads and its application in electrochemiluminescent immunoassay for CEA

PONE-D-25-47429R1

Dear Dr. Tang,

We’re pleased to inform you that your manuscript has been judged scientifically suitable for publication and will be formally accepted for publication once it meets all outstanding technical requirements.

Kind regards,

Shengwei Sun, Ph.D.

Academic Editor

PLOS ONE

Additional Editor Comments (optional):

Reviewers' comments:

Reviewer's Responses to Questions

**Comments to the Author**

Reviewer #1: (No Response)

2. Is the manuscript technically sound, and do the data support the conclusions?

Reviewer #1: (No Response)

3. Has the statistical analysis been performed appropriately and rigorously?

Reviewer #1: (No Response)

4. Have the authors made all data underlying the findings in their manuscript fully available?

Reviewer #1: (No Response)

5. Is the manuscript presented in an intelligible fashion and written in standard English?

Reviewer #1: (No Response)

Reviewer #1: (No Response)

**Do you want your identity to be public for this peer review?** For information about this choice, including consent withdrawal, please see our Privacy Policy

Reviewer #1: **Yes: ** Ahmed Al-Amiery

---

## [Editor Report · Acceptance letter]

PONE-D-25-47429R1

PLOS ONE

Dear Dr. Tang,

I'm pleased to inform you that your manuscript has been deemed suitable for publication in PLOS ONE. Congratulations! Your manuscript is now being handed over to our production team.

Kind regards,

on behalf of

Dr. Shengwei Sun

Academic Editor

PLOS ONE